# Exploring Independent and Cumulative Effects of Adverse Childhood Experiences on PTSD and CPTSD a Study in Ugandan Adolescents

**DOI:** 10.3390/children11050517

**Published:** 2024-04-25

**Authors:** Paulo Ferrajão, Francisco Frias, Ask Elklit

**Affiliations:** 1Faculdade de Ciências da Saúde, Universidade Europeia, Quinta do Bom Nome, Estrada da Correia 53, 1500-210 Lisbon, Portugal; fjfrias.92@gmail.com; 2CIDESD—Research Center in Sports Sciences, Health Sciences and Human Development, Universidade de Trás-os-Montes e Alto Douro, 5000-801 Vila Real, Portugal; 3National Center for Psychotraumatology, University of Southern Denmark, 5230 Odense, Denmark; aelklit@health.sdu.dk

**Keywords:** PTSD, complex PTSD, comorbidity, adverse childhood experiences, adolescence

## Abstract

Exposure to adverse childhood experiences (ACEs) is related to higher morbidity and mortality among adolescents. The present study analyzed the independent and cumulative effects of ACE exposure on the likelihood of PTSD and a CPSTD diagnosis in Ugandan adolescents. A sample of 401 schoolchildren participated in the study. The primary aim was to collect information on ACEs, PTSD, CPTSD, and attachment styles among adolescents living in different countries. It was found that exposure to 2–3 ACEs and exposure to 4–5 ACEs were significantly associated with PTSD diagnosis, while exposure to sexual abuse, bullying, threats of violence, and near-drowning were significantly related to CPTSD diagnosis. Fearful attachment style was significantly associated with PTSD diagnosis. The results propose that biological, psychological, and social issues interact and contribute to the differential prevalence of ACE, attachment styles, and PTSD/CPTSD. This study underscores the importance of addressing childhood-averse and traumatic experiences as a public health priority in Uganda.

## 1. Introduction

Adverse childhood experiences (ACEs) are developmental experiences occurring before age 18 that are not common in normative child development and frequently overwhelm a typical child’s regular coping skills [1]. These often comprise several types of violence and threat exposure (e.g., physical abuse, sexual abuse, bullying, or criminality), as well as different types of exposure to deprivation and loss (e.g., parental death, absence, or neglect) [2]. These events should be viewed as uncontrolled occurrences as they happen when the child or adolescent is not in a position of control, resulting in high distress levels for the victims [1].

Early and mid-2000s trauma research, specifically on ACEs and their relationship with psychological variables, focused primarily on single trauma-type exposure. Prior research has indicated that exposure to single-type ACE, such as child sexual abuse, is associated with mental health disorders, such as depression, post-traumatic stress disorder (PTSD), suicide, sexual promiscuity, and victim-perpetrator cycle [3]. This approach led to a possible underestimation of the prevalence of the co-occurrence of different types of ACEs.

However, there is evidence that the co-occurrence of multiple forms of ACEs is more prevalent than exposure to a single ACE [4,5]. The multiple exposure to different adverse and/or traumatic experiences has been defined as polytraumatization [6]. Prior research has found that people exposed to multiple traumas have more mental health problems compared to people exposed to a single traumatic event [4,7].

One stage of development recognized for its susceptibility to experiencing multiple types of ACEs is adolescence [8,9]. The increased risk of exposure to ACEs may be due to the particular characteristics of this period of development. Specifically, the biopsychosocial changes in adolescents, namely changes in their relationships with parents and peers involving a decrease in time spent with the primary caregivers and an increase in reliance on peers for social support and intimacy [10], may increase the occurrence of risky behaviors, such as perilous sexual behaviors and/or substance abuse, which may explain the vulnerability for polytraumatization among many adolescents [9].

### 1.1. Posttraumatic Stress Disorder (PTSD) and Complex Posttraumatic Stress Disorder (CPTSD)

As mentioned above, exposure to ACEs is associated with the development of various psychological disorders, namely PTSD and complex post-traumatic stress disorder (CPTSD). According to the International Classification of Diseases-11 (ICD-11), PTSD is characterized by three symptom clusters: trauma re-experience, avoidance of reminders of trauma, and sense of threat. CPTSD includes the three symptom clusters that characterize PTSD, but also three supplementary symptom clusters that indicate Disturbances in Self-Organization (DSO): affective dysregulation, negative self-concept, and disturbances in relationships [11].

In the fifth edition of the Diagnostic and Statistical Manual of Mental Disorders (DSM-5), CPTSD is not included CPTSD as a post-traumatic disorder by assuming that the addition of a fourth cluster to the PTSD diagnostic and the addition of a dissociative subtype would cover the full spectrum of disorders associated with traumatic experiences. The ICD-11 classifies PTSD and CPTSD as mutually exclusive disorders. Meanwhile, several studies conducted on different samples have found that PTSD and CPTSD are distinct disorders, providing evidence for the construct validity of CPTSD [12,13,14].

The different conceptualizations of ACEs exposure (i.e., single type vs. cumulative exposure) have led researchers to take different approaches to operationalizing the concept of polytraumatization. These are the cumulative, hierarchical, and categorical approaches. According to the hierarchical approach, specific types of ACEs are stronger predictors of negative mental health outcomes, e.g., sexual abuse is a stronger predictor for PTSD diagnosis than divorce [3,15]. The cumulative approach, which is used most often in psychotraumatology research, indicates that individuals with greater exposure to different types of ACEs are more likely to report poorer mental health [16]. Finally, the categorical approach posits that polytraumatization is best operationalized via different classes of exposure to ACEs, that is, individuals that exceed an exposure threshold (e.g., four or more types of ACEs are considered polytraumatized individuals and are more prone to report poorer mental health [5,17]. Previous research with either categorical or cumulative approaches used indexes of ACEs by summing up the number of events experienced and analyzing its cumulative effect on different biopsychosocial outcomes, which found that the probability of having a PTSD diagnosis increased gradually with the increase of ACEs [17,18].

The association between ACEs and PTSD and/or CPTSD varies according to the methodological approach, i.e., whether independent or cumulative effects are analyzed. Although the cumulative approach is the most widely used, the analysis of the effect of different kinds of ACEs on PTSD and CPTSD should also be considered. Some single-type ACE may be associated, or not, with PTSD and/or CPTSD. To afford a better comprehension of the relationship between ACEs and both PTSD and CPTSD, it is crucial to analyze the effects of both cumulative and different ACEs on both PTSD and CPTSD among adolescents.

Several studies that analyzed the effect of cumulative exposure to ACEs have found that individuals with higher exposure to ACEs had higher PTSD symptom levels compared to individuals with lower levels of exposure or exposure to a single ACE [19,20,21]. It was also observed that individuals exposed to less than four ACEs had a higher risk of PTSD than CPTSD, whereas individuals exposed to six or more ACEs had a higher risk of CPTSD [22]. However, it was found that exposure to ACEs had only an indirect effect on the development of PTSD and CPTSD symptoms, and the indirect effect was greater for PTSD symptoms [23].

As for the independent effect of ACEs, it seems that different types of ACEs have distinct associations with both PTSD and CPTSD. However, these associations are not uniform in different studies. For instance, it was found that witnessing violence and physical neglect increased the probability of higher levels of PTSD symptoms, while sexual abuse and emotional neglect were not linked to PTSD symptoms [24]. It was observed sexual abuse had a stronger association with PTSD and DSO symptoms compared to physical neglect [25]. It was also found that childhood sexual abuse and childhood physical assault significantly amplified the probability of developing CPTSD in comparison with PTSD, whereas near drowning was more likely to be associated with PTSD [25].

### 1.2. Low- and Lower-Middle-Income Countries (LALMIC)

Most studies on this topic have been performed in adult populations from Western countries, which hinders the generalization of the findings to other populations, such as adolescents from low- and lower-middle-income countries (LALMIC). Research on ACEs in samples from LALMIC is scarce in comparison with other countries. LALMIC diverges from other countries in general living conditions, accessibility to healthcare, and social contexts due to higher poverty levels and social instability [26].

This socioeconomic context may originate in a higher prevalence and higher prevalence of exposure to adverse or traumatic events compared to Western populations [26,27,28]. This risk is notably higher in adolescents living in LALMIC. However, there is a lack of research on the between exposure to ACEs and psychological disorders among samples of African countries. The case of Uganda is one such country, and our study aimed to analyze the relationship between exposure to ACEs with both PTSD and CPTSD.

Uganda has a recent history of civil war and uprisings with widespread violation of children’s rights from exposure to traumatic events such as direct involvement in the armed conflict as child soldiers, child labor, and subsequent risks for sexual abuse or human trafficking [29,30]. In this context, it seems very likely that many youth have been exposed to multiple types of ACEs [31,32]. Therefore, the study of exposure to such experiences among Ugandan adolescents is vital for developing interventions and prevention policies tailored to this population.

Uganda has nine million adolescents, totaling 25% of the population. However, numerous are hindered by poverty, severe illnesses, early pregnancy, gender-based violence, and low secondary education enrolment [33]. In these, adolescent girls are especially vulnerable. For example, it has been observed that bullying and caning are common occurrences among Ugandan adolescent girls due to the fact that they suffer more corporal punishment from teachers and educators compared to boys [34].

These traumatic experiences have a damaging effect on the adolescents’ mental health. Prior research has analyzed the relationship between ACEs exposure and diagnosis of PTSD and CPTSD, including in samples of LALMIC. It was found that between 0.6–23.8% were exposed to one ACE, between 14.8–30.8% were exposed to two or three ACEs, and between 45.4–84.5% were exposed to 4 or more ACEs in Ugandan samples [35,36]. In Ugandan children and adolescents, the prevalence of PTSD ranged between 13% and 67.5% [37,38,39,40,41,42].

Regarding the relationship between ACEs exposure and PTSD, two studies conducted with samples from Uganda noticed a dose–response relationship between exposure to ACEs and PTSD symptoms. It was found that the total number of ACEs was positively associated with PTSD scores [41]. In another study, it was observed that in comparison to individuals who were exposed to 0–3 ACEs, individuals exposed to 4–7 events were two and a half times more likely to develop PTSD, while individuals exposed to 8–11 and 12–16 events had a greater likelihood by four and a half and six and a half times, respectively [42]. In a performed in a sample of young Congolese girls, it was found that the probability of having a PTSD diagnosis increased gradually with the increase of ACEs experienced, while the exposure to a single ACE, such as abuse with a firearm and rape, considerably increased the probability of PTSD by almost five times [37].

### 1.3. Psychosocial Variables

The discrepancies in research findings on the association between ACE exposure and psychological disorders have prompted researchers to consider the involvement of other variables in explaining the development of trauma-related disorders. As not every individual who has experienced exposure to multiple ACEs develops PTSD or CPTSD, this indicates the presence of additional factors influencing the relationship between exposure to ACEs and these conditions.

There appear to be differences between females and males in the likelihood of developing PTSD and CPTSD symptoms. Previous studies found differences between both sexes in the exposure to specific traumatic events, with females being more likely to be exposed to childhood sexual abuse [33,43,44,45]. In turn, exposure to other ACEs, such as child sexual abuse, can be exacerbated by the family structure in which the child or adolescent is inserted. Specifically, it was observed that most sexually abused girls lived in single-parent households [46].

Another variable that seems to contribute to a higher risk of PTSD and CPTSD symptoms because of exposure to ACEs is attachment style. According to attachment theory, humans have an inborn drive to seek proximity and comfort from attachment figures, particularly in times of distress or threat [47]. The primary experiences with caregivers shape an individual’s ability to form social connections throughout life [48]. The quality of these early attachment experiences sets the stage for the formation of internal working models of self and others (IWM) [49].

The IWM are mental frameworks or schemas that shape individuals’ expectations, perceptions, and behaviors towards attachment figures and close relationships. These models are formed through a process of assimilating and organizing attachment-related information and serve as templates for understanding and navigating social interactions throughout the lifespan. IWM includes beliefs about oneself (e.g., self-worth, lovability) and others (e.g., trustworthiness, availability) and can influence emotional experiences, relational patterns, and coping strategies in relationships [50].

The attachment system can be activated by physical separation and/or danger. Two primary strategies are used to regulate attachment-related distress: anxious and avoidant attachment. Attachment anxiety involves the hyperactivation of the attachment system in response to perceived threats and/or rejection, leading to a preoccupation with attachment-related concerns, heightened vigilance, and seeking proximity and reassurance from attachment figures. Attachment avoidance involves the deactivation of the attachment system by minimizing the importance of close relationships, avoiding emotional intimacy, and emphasizing self-reliance and independence [50].

According to this theoretical framework, attachment styles are conceptualized as patterns of thoughts, emotions, and behaviors that individuals use to regulate attachment-related anxiety and avoidance in close relationships [51]. Individuals with secure attachment styles possess a positive view of themselves and others, are comfortable with intimacy, and seek emotional closeness with others. Individuals with preoccupied attachment styles present a high level of attachment-related anxiety, being strongly concerned with the availability and responsiveness of their attachment figures, which results in reliance on others for validation and reassurance. Individuals with a dismissive attachment mainly adopt avoidant strategies, viewing intimacy as a threat to their independence, which results in suppression of their emotions and rejection of seeking support from others. Individuals with a fearful attachment style exhibit both anxious and avoidant behaviors, presenting inconsistent and often contradictory behaviors in close relationships [50].

Research on trauma exposure, attachment styles, and mental health consequences, specifically PTSD and CPTSD, has produced results that corroborate the existence of a relationship between these variables. A meta-analytic study reported that both anxious and fearful attachment styles were positively related to PTSD symptoms, whereas secure attachment was negatively linked to PTSD symptoms [51]. Dismissive attachment was not associated with PTSD symptoms, which may be due to the lower reporting of psychological distress that is common in individuals with a dismissing attachment style [50,52].

In a systematic review covering the association between different types of traumatic events and attachment styles, the authors found that interpersonal trauma (i.e., traumatic events involving other people) had a greater association with insecure attachment styles than non-interpersonal trauma [52]. In a study that analyzed the link between attachment styles with PTSD and DSO symptoms, it was found that secure and fearful attachments were negatively and positively, respectively, linked to DSO symptoms, and dismissive attachment was positively associated with both PTSD and DSO symptoms [53]. Although there is very little research focused on attachment and mental health outcomes, it was found that secure parental attachment was a protective factor against the development of anxiety and depression symptoms among Ugandan adolescents [54,55].

### 1.4. Purpose of the Study

Most studies usually analyze either the independent or cumulative effect of ACE exposure on mental health outcomes. There is a lack of research that has analyzed the effect of both approaches in the same study. It analyzed the impact of both independent and cumulative exposure to ACEs on both PTSD and CPTSD in a sample of Ugandan adolescents.

Moreover, some variables, such as sex and attachment style, have been proposed to influence the association between exposure to ACEs and mental health outcomes [44,46,50,52]. These variables seem to be key factors in the development of psychopathology because of exposure to ACEs. In this study, the effect of sex, attachment styles, and living arrangements on both PTSD and CPTSD will be analyzed.

This study aims to provide a more comprehensive understanding of the associations between exposure to ACEs, attachment, PTSD, and CPTSD in Ugandan adolescents. Specifically, we intend to: (a) identify the prevalence of exposure to ACEs, PTSD, CPTSD, and attachment styles; (b) analyze the independent effects of each ACE on PTSD and CPTSD diagnosis; (c) analyze the cumulative effect of ACEs exposure on PTSD and CPTSD diagnosis; and, (d) analyze the effect of attachment style, sex, and living arrangements on PTSD and CPTSD diagnosis.

## 2. Method

### 2.1. Sampling, Study Design, and Data Collection

A sample of 401 schoolchildren was selected and invited to take part in the study. All the participants were enrolled in the eighth grade. Table 1 presents the sample’s characteristics. The mean age was around 15.9 years, ranging from 13 to 19 years. The percentage of females and males was very similar. Most of the participants lived with both parents, many with one parent, and a considerable number had other arrangements.

Eight different boarding schools were chosen from three major cities in Uganda. Three of the schools were in Kampala, the capital of Uganda, which is a city located in central Uganda; three of the schools were in Mbarara, which is a city located in Western Uganda; the remaining two schools were in Jinja, which is in Eastern Uganda. Boarding schools were introduced into Uganda’s education system with the aim of ensuring the universalization of secondary education. They are mostly aimed at disadvantaged children from underprivileged regions and/or poorer households without due payment. The schools selected for this study were seen as typical schools in the Ugandan school system following the suggestion of a Danish psychologist who had worked in Uganda for 30 years as a school psychologist and consultant and who had a wide acquaintance with the Ugandan school system. All schools but one (one boarding school in Jinja) completed the surveys within a ten-day period in August 2012. The last one did not join because it only had one student at the eighth-grade level.

The research protocol was reviewed and approved by the Institutional Review Board of the Ugandan National Council for Science and Technology. Following, the aim of the study and the confidentiality principles were explained to the headmasters of the selected schools. Because English is the official language of Uganda, as well as the Ugandan school system, the questionnaires were in English.

Passive consent was applied, which is a common procedure in school studies in the LALMIC. We informed the parents about the objectives and procedures of the study, and they could inform the research team about their decision to refuse their child’s participation in the study. In Uganda, the parents have confidence in the school system, and the teachers are in *loco parents*, i.e., they are offered the role of acting in the best interests of their children. A short introduction was given before handing out the questionnaires to the participants, explaining the objective of the study, the option of refusing to participate, the privacy principles, and the procedure of delivering the questionnaire in one envelope and sealing it up in front of the students. Participants gave their consent to take part in the study, and all students agreed to participate. Participants were given the opportunity to speak to a psychologist after the data collection if they felt emotionally upset or embarrassed by the topics covered in the questionnaires.

Around 15–20 min was spent on introduction and explanation before the participants were invited to fill out the questionnaires. Participants took around one and a half hours to answer the questionnaires. Headmasters of all schools were asked if they could provide some teachers to assist in answering and explaining the questions that the researcher was not able to, such as language difficulties, and indeed, the teachers were very helpful with this. The students were given pens and calculators as a gesture of gratitude for their participation in the study.

### 2.2. Measures

#### 2.2.1. Sociodemographic Data

Participants were asked to provide information on their sex (male and female), age, and current living arrangements (living with both parents, living with one of the parents, and other living arrangements).

#### 2.2.2. Adverse Childhood Experiences

A questionnaire with a list of 19 life-threatening experiences (e.g., rape) and stressful family conditions (e.g., neglect) was presented to the participants, and they were asked if they had been exposed to these events. These events were selected from scholarly literature and clinical practice [56]. This measure has been used previously in studies among African adolescents [57]. Because of the similarity of some events, a list of 15 events was assembled (Table 2) and analyzed in the current study [58].

#### 2.2.3. Revised Adult Attachment Scale (RAAS)

The RAAS [59] was used to measure participants’ attachment styles. This self-report measure comprises 18 items that are scored on a 5-point Likert scale (from “not at all characteristic of me” = 1 to “very characteristic of me” = 5). The scale contains three dimensions: (a) closeness attachment, (b) dependency attachment, and (c) six anxious attachment. Each dimension includes six items. The index score of attachment anxiety orientation was calculated by adding the items of the anxious dimension. The index score of attachment avoidance orientation was calculated by adding the index scores of both closeness and dependency dimensions. The RAAS is a widely used measure for attachment dimensions and attachment styles among adolescents. It has been used in previous studies on samples of African adolescents [56]. The categorical attachment styles were distributed by values on the dimensions of anxiety and avoidance: participants with secure attachment had low scores on both anxiety and avoidance dimensions; participants with a preoccupied attachment had high scores on the anxiety dimension and low scores on the avoidance dimension; participants with a dismissing style had low scores on the anxiety dimension and high scores on the avoidance dimension; participants with a fearful attachment had high scores on both anxiety and avoidance dimensions. High is defined as being above the midpoint on the 5-point scale, and low is equal to or below the midpoint. The reliability of the attachment anxiety scale (α = 0.78) and the attachment avoidance scale (α = 0.76) were satisfactory.

#### 2.2.4. PTSD Item Set

PTSD symptoms were assessed using six items selected [60] from the Harvard Trauma Questionnaire: Part IV (HTQ-IV) [61]. These items are answered on a 4-point Likert scale (from “not present” = 1 to “very often present” = 4). Table 3 lists the items representing PTSD. PTSD diagnosis was performed using ICD-11 criteria. A diagnosis of PTSD requires the endorsement of one of two symptoms from the symptom clusters of (1) re-experiencing the traumatic events, (2) avoidance, and (3) a sense of current threat, in addition to the endorsement of one or more indicators of functional impairment. Endorsement of a symptom or functional impairment item is defined as a score equal to or higher than 2. The reliability of the scale (α = 0.78) was satisfactory.

#### 2.2.5. CPTSD Item Set

CPTSD symptoms were assessed using six items: one item was retrieved from the HTQ-IV, and five items were retrieved from the Trauma Symptom Checklist (TSC) [62]. The TSC items are answered on a 4-point Likert scale (from “never” = 0 to “very often” = 3). These items measure the CPTSD clusters (affective dysregulation, negative self-concept, and disturbances in relationships) [60]. Table 3 lists the items representing CPTSD. A diagnosis of CPTSD requires the endorsement of one of two symptoms from each of the three PTSD symptoms clusters and one of two symptoms from each of the three Disturbances in Self-Organization (DSO) clusters: (1) affective dysregulation, (2) negative self-concept, and (3) disturbances in relationships. The reliability of the scale (α = 0.79) was satisfactory.

### 2.3. Data Analysis

Data analysis was conducted using the IBM SPSS Statistics for Windows (version 29). Descriptive analyses were conducted to analyze sample characteristics. The prevalence of ACEs was analyzed, and a sequence of Chi-square tests was performed to compare both sexes on exposure to different types of ACEs. Subsequently, two multivariable logistic regression analyses were performed to analyze the independent and cumulative impacts of ACEs on PTSD and CPTSD diagnosis. A multivariable logistic regression model is valuable when testing the effects of independent variables on a nominal dependent variable, here, diagnostic criteria or not, for PTSD and/or CPTSD. The sample size was large enough to conduct multinomial logistic regression since multivariable logistic regression requires a minimum of 10 cases per independent variable. The Nagelkerke R^2^ [63] was chosen to obtain the R^2^ in multinomial logistic regression because it adjusts the Cox-Snell R^2^ [64] by dividing Cox-Snell R^2^ by its upper bound, for a more intuitive interpretation of R2, such as R^2^ in the linear regression model.

The first multivariable logistic regression model included the following independent variables: individual ACEs items, sex (female or not), living arrangements (child lives with both parents or not), secure attachment (yes or no), preoccupied attachment (yes or no), dismissing attachment (yes or no), and fearful attachment (yes or no), PTSD diagnosis (yes or no), and CPTSD diagnosis (yes or no). The second regression model had a cumulative index of ACEs (i.e., categories of ACEs) instead of individual ACEs items and the remaining variables that were included in the first regression model. The odds ratios indicate the expected increase/decrease in the likelihood of scoring positively on a given variable compared with the reference group for each independent variable.

## 3. Results

### 3.1. Prevalence of ACEs

The most reported event was the death of someone close, followed by physical violence, bullying and threats of violence, and serious illness, which were reported by more than half of the participants. The least prevalent was pregnancy/abortion, followed by attempted suicide and sexual abuse. A series of Chi-square analyses were conducted to analyze the associations between sex and the prevalence of ACEs. It was found that there were significant Chi-square statistics between both sexes on sexual abuse, childhood neglect, and absence of a parent. Specifically, the proportion of exposure to those ACEs was higher in females compared to males (Table 4).

### 3.2. Attachment Styles, and PTSD and CPTSD Diagnosis

As can be seen in Table 4, secure attachment had the highest proportion, followed by preoccupied style. The least prevalent attachment style was dismissing attachment. Around a quarter of the participants met the criteria for a diagnosis of PTSD, and approximately a third of the participants met the criteria for a diagnosis of CPTSD. There were significant Chi-square statistics between both sexes on PTSD diagnosis. Specifically, the proportion of participants with a PTSD diagnosis was higher in females compared to males (Table 5).

### 3.3. Independent and Cumulative Effects of ACEs on PTSD Diagnosis

It is performed first multivariable logistic regression model to examine the relationship between independent ACEs and PTSD diagnosis. Table 5 presents the odd ratios and Confidence Intervals associated with each predictor. Sex was the only sociodemographic characteristic that was significantly associated with PTSD diagnosis. Females were twice as likely to have a diagnosis of PTSD. Regarding the categorical attachment styles, only the fearful attachment style was significantly associated with PTSD diagnosis. Ugandan adolescents with a fearful attachment style were three and a half times more likely to have a PTSD diagnosis. Regarding the independent ACEs, it was observed that exposure to physical violence, witnessing other people injured or killed, and attempted suicide were significantly associated with PTSD diagnosis. While participants exposed to the latter two experiences were nearly two and a half times more likely to have a PTSD diagnosis, participants exposed to the former were twice as likely to have a PTSD diagnosis.

A second multivariable logistic regression model was performed to analyze the relationship between cumulative exposure to ACEs and PTSD diagnosis. Table 6 presents the Odd Ratios and Confidence Intervals associated with each predictor. Likewise, sex was the only sociodemographic characteristic that was significantly associated with PTSD diagnosis, presenting the same probability. Regarding the categorical attachment styles, the fearful attachment style was significantly associated with PTSD diagnosis, with the same probability of having a PTSD diagnosis. Regarding the cumulative exposure to ACEs, exposure to 2–3 ACEs and exposure to 4–5 ACEs were significantly associated with PTSD diagnosis. Ugandan adolescents exposed to 2–3 ACEs were almost nine times more likely to present a PTSD diagnosis, and adolescents exposed to 4–5 ACEs were twice as likely to have a diagnosis of PTSD.

### 3.4. Independent and Cumulative Effects of ACEs on CPTSD Diagnosis

First, a multivariable logistic regression model was performed to examine the relationship between independent ACEs and CPTSD diagnosis. Table 7 presents the Odd Ratios and Confidence Intervals associated with each predictor. None of the sociodemographic variables included in the model were associated with CPTSD diagnosis. Regarding the categorical attachment styles, only dismissing attachment style was significantly associated with CPTSD diagnosis. Adolescents with a dismissing attachment style were twice as likely to have a CPTSD diagnosis. Regarding the independent ACEs, exposure to sexual abuse, bullying, threats of violence, and near-drowning were significantly associated with CPTSD diagnosis. Exposure to these ACEs increased the risk of having a CPTSD diagnosis twofold.

Finally, a multivariable logistic regression model was conducted to examine the relationship between cumulative exposure to ACEs and CPTSD diagnosis. Table 8 presents the Odd Ratios and Confidence Intervals associated with each predictor. None of the sociodemographic variables were associated with CPTSD diagnosis. Moreover, none of the categorical attachment styles were associated with CPTSD diagnosis. Regarding the cumulative exposure to ACEs, only exposure to 4–5 ACEs and exposure to six or more ACEs were significantly associated with CPTSD diagnosis. Ugandan adolescents exposed to 4–5 ACEs were approximately two and a half times more likely to have a CPTSD diagnosis, and adolescents exposed to six or more ACEs were one and a half times more likely to be diagnosed with CPTSD.

## 4. Discussion

The main purpose of the present study was to ascertain both the independent and cumulative effects of ACEs on the likelihood of PTSD and a CPSTD diagnosis in Ugandan adolescents. We also considered sex, living arrangements, and attachment styles as predictors in performing multivariable logistic regression analyses for independent and cumulative effects of ACEs. Results of the current study suggest rates of PTSD like other studies with African and South Asian LALMIC samples [38,40,41,42]. The prevalence of CPTSD was like other studies in the African sample [62]. These results suggest that Ugandan adolescents have a higher risk of presenting PTSD and CPTSD compared to other samples from Western countries [65,66].

### 4.1. Independent Effect of ACEs on PTSD/CPTSD

The results obtained in the multivariable logistic regression for the independent effect of ACEs on PTSD and CPTSD are not consistent with previous literature. However, there were some differences in the types of ACEs that increased the likelihood of either PTSD or CPTSD diagnosis. It was observed that witnessing other people being injured or killed, physical violence, and attempted suicide significantly increased twofold the likelihood of a PTSD diagnosis. These findings suggest that life-threatening experiences, suffered or witnessed, involving physical violence or horrific events, such as attempted suicide, increase the likelihood of developing PTSD in Ugandan adolescents [21]. Considering the high prevalence of direct exposure or testimony of physical violence, it seems to be a community factor that contributes to the development of PTSD in these adolescents [33,38].

On the other hand, having experienced sexual abuse, bullying, threats of violence, and near-drowning increased the likelihood of a CPTSD diagnosis by twofold. These results are in accordance with literature that argues that a diagnosis of CPTSD is more likely because of exposure to events that usually involve extended or recurrent occurrences from which escape is problematic or unbearable, such as sexual abuse and bullying [11,25]. Both experiences may involve betrayal by relatives or peers, resulting in a complex interplay of strong emotions and maladaptive coping mechanisms that exacerbate the consequences of the trauma experienced by the individual [67,68]. The high prevalence of drowning found in the sample (almost a third of the participants) may reflect the co-occurrence of risk-taking behaviors typical of adolescents, which puts adolescents at a greater risk for exposure to other ACEs and increases the risk of CPTSD [69,70].

### 4.2. The Cumulative Effect of ACEs on PTSD/CPTSD

The results of the current study do not indicate a linear dose–response relationship between exposure to ACEs and PTSD and/or CPTSD found in previous studies [71]. Instead, our results indicate that cumulative ACE exposure may be more strongly associated with either PTSD or CPTSD, depending on the number of ACEs a person has been exposed [22,66].

It was observed that exposure to 2–3 ACEs increased the risk of PTSD diagnosis by almost nine times, while exposure to 4–5 ACEs increased the likelihood of PTSD diagnosis by twofold. However, being exposed to six or more ACEs did not increase the odds of receiving a PTSD diagnosis. As for the cumulative effect of ACEs on CPTSD, only exposure to 4–5 ACEs and exposure to six or more ACEs were significantly linked to CPTSD diagnosis. It was observed that adolescents exposed to 4–5 ACEs were approximately two and a half times more likely to have a CPTSD diagnosis, and those exposed to six or more ACEs were one and a half times more likely to be diagnosed with CPTSD.

The current results are in line with previous empirical studies indicating that adolescents exposed to multiple ACEs present a higher risk of post-traumatic disorders compared to adolescents exposed to a single ACE [17,20]. The current results also propose that adolescents exposed to less than four ACEs (between 2 and 3) had an extremely high probability of being diagnosed with PTSD, and exposure to higher levels of ACEs increases the likelihood of a diagnosis of CPTSD [22].

### 4.3. Sex, Living Arrangements and Attachment Style

It was observed that being female increased the likelihood of a PTSD diagnosis twofold, which may be related to differences in exposure to ACEs in both sexes [72]. In accordance with the literature, the greater likelihood of PTSD diagnosis in females may be due to greater exposure to sexual abuse, childhood neglect, and various forms of gender-based violence and discrimination compared to males [44,73,74].

Regarding the distribution of the attachment styles, only one-third of the adolescents had a secure attachment style. In the non-secure attachment styles, almost a third of the participants had a preoccupied attachment, a fifth had a fearful attachment, and a smaller proportion had a dismissing attachment. The high level of exposure to ACEs probably contributes to the high proportion of adolescents with non-secure attachment styles [54,75]. High exposure of teenagers to the death of someone close and physical violence could intensify the possession of negative models of self and others associated with insecure attachment anxiety, which amplifies adolescents’ concerns about the unavailability of attachment figures [76].

Our results indicate that having a fearful attachment style amplified the odds of PTSD diagnosis by almost three and a half, but none of the attachment styles had a statistically significant effect on the likelihood of a CPTSD diagnosis. The present results follow previous evidence of a higher risk of PTSD in adolescents with a fearful attachment system [51,77,78]. It seems that Ugandan adolescents with negative models of the self and others who developed inconsistent and contradictory relationships with attachment figures are at greater risk of presenting PTSD [50].

Contrary to expectations, fearful attachment was not associated with CPTSD diagnosis. Other factors may have a more significant impact on the relationship between ACE exposure and CPTSD in the context of Ugandan adolescents. According to the cascade model of CPTSD, the development of CPTSD depends on multiple tiers of mediated associations between variables [79]. It can be proposed that other psychosocial factors, such as disclosure of trauma, social acknowledgment, and/or perceived social support, may mediate the association between attachment systems and CPTSD among Ugandan adolescents.

Regarding the model fit for the models, Model 1 showed a moderate relationship between the predictors and the outcome, while the remaining models showed a weak relationship between the predictors and the outcomes. These results suggest that other variables, such as the initial age of exposure to ACE or coping styles, may be associated with the development of PTSD and CPTSD. Future studies could, therefore, include these variables as predictors of both disorders.

### 4.4. Limitations

The current study had some limitations. First, the cross-sectional design of the study disallows inference of causality. Second, the study variables were assessed through self-report measures, which entails the risk of a reporting bias. Third, a report of exposure to ACEs was performed retrospectively, which can be biased by memory issues. Fourth, the assessment of exposure to ACEs did not account for the reoccurrence of specific events. Fifth, the data were collected in 2012. Thus, there may have been changes in the type and prevalence of ACEs to which adolescents are exposed nowadays. Sixth, a convenience sample was employed in this study. Therefore, it is important to acknowledge the limitations in generalizing our findings, as our sample was drawn from specific locations and contexts in Uganda and may not be representative of the broader population. Seventh, despite the existence of differences between both sexes in the proportion of exposure to some ACEs and in the diagnosis of PTSD, the present study did not test the differences in the models between both sexes. It is recommended that future studies analyze the models clustered by sex. Moreover, it is recommended that future studies explore the intracultural differences within LALMIC. It is suggested that future studies analyze the independent and cumulative effects of exposure to ACEs on the mental health of adolescents in other cultural contexts.

## 5. Conclusions

Several trauma treatment models have been developed for children, focusing on addressing specific types of traumatic events. However, there is a necessity for trauma interventions that address exposure to multiple types of traumatic events. This is particularly relevant in LALMIC, where a significant proportion of the global youth population lives. Unfortunately, there is a lack of research addressing exposure to multiple ACEs and mental health in LALMIC, such as in Uganda. The present study contributes to the understanding of ACE exposure in LALMIC and provides a foundation for implementing community-based initiatives in these regions.

The current results suggest that exposure to ACEs has a significant adverse impact on Ugandan adolescents’ mental health, namely a strong association between exposure to ACEs and the development of PTSD and CPTSD. Exposure to multiple ACEs can have long-lasting effects on mental health outcomes, particularly in LALMIC, where resources for mental health care are limited. The high prevalence of ACEs in LALMIC adolescents highlights the urgent need for interventions aimed at preventing and addressing childhood trauma.

The different types of ACEs and their independent effects on the development of PTSD and CPTSD do not imply a causal relationship. Instead, it suggests that individuals who have encountered specific types of ACEs may be at a heightened risk of developing PTSD and/or CPTSD. This finding hints at the existence of patterns of circumstances that amplify vulnerability to these disorders, as well as the likelihood and frequency of experiencing these events.

Overall, this study underscores the importance of addressing childhood trauma as a public health priority in Uganda. Designing effective prevention initiatives that address multiple exposures to ACEs requires a comprehensive approach that meets the specific needs of adolescents. Furthermore, it is important for healthcare professionals to prioritize the assessment of the entire traumatic background rather than solely focusing on the most recognized traumatic experiences. Moreover, it was observed that attachment plays a significant role in the development of PTSD. The high prevalence of non-secure attachment styles may reflect the impact of ACEs on the development of attachment patterns and subsequent psychological well-being. Thus, one of the objectives of the intervention with these adolescents will be developing more positive IWM of themselves and others, which could contribute to a better adjustment to exposure to multiple types of ACEs. Overall, these findings emphasize the importance of addressing social determinants of health and promoting supportive environments, including family, to prevent and mitigate the impact of ACEs and facilitate recovery from trauma.

## Figures and Tables

**Table 1 children-11-00517-t001:** Sample demographic characteristics.

Sociodemographic Variables	Total (N = 401)
**Age**	
13 years	7 (1.7%)
14 years	50 (12.5%)
15 years	89 (22.2%)
16 years	139 (34.7%)
17 years	81 (20.2%)
18 years	27 (6.7%)
19 years	8 (2.0%)
Mean	15.9 (SD = 1.2)
**Sex**	
Male	198 (49.4%)
Female	203 (50.6%)
**Living with**	
Both parents	170 (42.4%)
One of their parents	142 (35.4%)
Other arrangements (uncles, siblings, grandparents, or other relatives)	89 (22.2%)

**Table 2 children-11-00517-t002:** List of original adverse childhood experiences and grouped adverse childhood experiences.

Original Adverse Childhood Experiences	Grouped Adverse Childhood Experiences
Traffic accident	Serious accidents
Other serious accidents	
Physical assault	Physical violence
Physical abuse	
Rape	Sexual abuse
Sexual abuse	
Witnessed other people injured or killed	Witnessed other people injured or killed
Came close to being injured or killed	Came close to being injured or killed
Humiliation or persecution (bullying)	Bullying and threats of violence
Threats of violence	
Near-drowning	Near-drowning
Attempted suicide	Attempted suicide
Robbery/theft	Robbery/theft
Pregnancy/abortion	Pregnancy/abortion
Serious illness	Serious illness
Death of someone close	Death of someone close
Divorce	Divorce
Severe childhood neglect	Childhood neglect
Absence of a parent	Absence of a parent

**Table 3 children-11-00517-t003:** Items representing PTSD and Complex PTSD symptoms.

Cluster	Test Items
PTSD symptoms	HTQ 2. Feeling as though the event is happening again HTQ 3. Recurrent nightmares HTQ 6. Being jumpy or easily startled HTQ 9. Feeling on guard HTQ 11. Avoiding activities that remind you of the traumatic or hurtful event HTQ 15. Avoiding thoughts or feelings associated with the traumatic or hurtful events
CPTSD symptoms	TSC 6. Feeling isolated from other people TSC 14. Crying easily
	TSC 16. Temper outburst that you could not control
	TSC 28. Feelings of inferiority or insecurity
	TSC 29. Blaming yourself
	HTQ 27. Feeling that you have no one to rely upon

**Table 4 children-11-00517-t004:** Chi-Square analyses between group sex of the participants and exposure to adverse childhood experiences, attachment styles, and PTSD and CPTSD diagnosis.

Variables	Females (n = 203) Count (%)	Males (n = 198) Count (%)	Full Sample (N = 401) Count (%)	χ^2^
Adverse childhood experiences				
Serious accidents	104 (51.2%)	92 (46.5%)	196 (48.9%)	0.91
Physical violence	144 (70.9%)	127 (64.1%)	271 (67.6%)	2.12
Sexual abuse	50 (24.6%)	26 (13.1%)	76 (19.0%)	8.63 **
Witnessed other people injured or killed	86 (42.4%)	91 (46.0%)	177 (44.1%)	0.53
Came close to being injured or killed	94 (46.3%)	92 (46.5%)	186 (46.4%)	0.01
Bullying and threats of violence	135 (66.5%)	136 (68.7%)	271 (67.6%)	0.22
Near-drowning	62 (30.5%)	62 (31.3%)	124 (30.9%)	0.03
Attempted suicide	42 (20.7%)	30 (15.2%)	72 (18.0%)	2.09
Robbery/theft	91 (44.8%)	85 (42.9%)	176 (43.9%)	0.15
Pregnancy/abortion	16 (7.9%)	10 (5.1%)	26 (6.5%)	1.33
Serious illness	138 (68.0%)	127 (64.1%)	265 (66.1%)	0.70
Death of someone close	160 (78.8%)	146 (73.7%)	306 (76.3%)	1.43
Divorce	61 (30.0%)	46 (23.2%)	107 (26.7%)	2.38
Childhood neglect	62 (32.5%)	41 (20.7%)	103 (25.7%)	5.08 *
Absence of a parent	105 (51.7%)	77 (38.9%)	182 (45.4%)	5.66 **
Attachment styles				
Secure	65 (32.0%)	70 (35.4%)	135 (33.7%)	0.50
Preoccupied	62 (30.5%)	59 (39.8%)	121 (30.2%)	0.03
Dismissing	29 (14.3%)	30 (15.2%)	59 (14.7%)	0.06
Fearful	47 (23.2%)	39 (19.7%)	86 (21.4%)	0.71
Diagnosis				
PTSD	60 (29.6%)	51 (25.8%)	111 (27.7%)	13.09 ***
CPTSD	74 (36.5%)	71 (35.9%)	145 (36.2%)	1.88

Note: * *p* < 0.05. ** *p* < 0.01. *** *p* < 0.001.

**Table 5 children-11-00517-t005:** Results of the multivariable logistic regression for PTSD: Independent ACEs.

Predictor	Odds Ratio	CI Interval
Sex (female)	2.14 ***	(1.34, 3.40)
Living arrangements (One parent or other arrangements)	1.15	(0.93, 2.52)
Secure	0.80	(0.41, 1.58)
Preoccupied	0.69	(0.34, 1.39)
Dismissing	1.74	(0.98, ~3.03)
Fearful	3.43 **	(1.56, 7.58)
Serious accidents	0.83	(0.52, 1.33)
Physical violence	2.13 **	(1.27, 3.58)
Sexual abuse	1.58	(0.82, 3.07)
Witnessed other people injured or killed	2.31 ***	(1.40, 3.80)
Came close to being injured or killed	1.00	(0.63, 1.63)
Bullying and threats of violence	1.06	(0.62, 1.80)
Near-drowning	1.26	(0.74, 2.14)
Attempted suicide	2.42 *	(1.21, 4.86)
Robbery/theft	1.02	(0.63, 1.63)
Pregnancy/abortion	0.46	(0.17, 1.25)
Serious illness	1.07	(0.65, 1.78)
Death of someone close	1.25	(0.74, 2.14)
Divorce	1.24	(0.68, 2.24)
Childhood neglect	0.92	(0.50, 1.68)
Absence of a parent	0.82	(0.48, 1.39)

Notes: Reference group = None, n = 401. Nagelkerke R^2^ = 0.23, * *p* < 0.05. ** *p* < 0.01. *** *p* < 0.001.

**Table 6 children-11-00517-t006:** Results of the multivariable logistic regression for PTSD: Cumulative ACEs.

Predictor	Odds Ratio	CI Interval
Sex (female)	2.04 **	(1.32, 3.15)
Living arrangements (One parent or other arrangements)	1.42	(0.91, 2.21)
Secure	0.82	(0.43, 1.55)
Preoccupied	0.65	(0.33, 1.26)
Dismissing	1.69	(0.96, 3.00)
Fearful	3.35 **	(1.56, 7.19)
ACE (1)	3.73	(0.76, 18.29)
ACE (2–3)	8.78 **	(2.12, 36.42)
ACE (4–5)	2.27 *	(1.17, 4.41)
ACE (≥6)	1.38	(0.79, 2.42)

Notes: n = 401. Nagelkerke R^2^ = 0.16. * *p* < 0.05. ** *p* < 0.01.

**Table 7 children-11-00517-t007:** Results of the multivariable logistic regression for CPTSD: Independent ACEs.

Predictor	Odds Ratio	CI Interval
Sex (female)	1.38	(0.90, 2.14)
Living arrangements (One parent or other arrangements)	1.11	(0.69, 1.80)
Secure (No)	1.39	(0.89, 2.17)
Preoccupied	1.10	(0.64, 1.89)
Dismissing	2.11 *	(1.08, 4.14)
Fearful	1.37	(0.76, 2.48)
Serious accidents	0.74	(0.47, 1.16)
Physical violence	1.59	(0.96, 2.62)
Sexual abuse	1.93 *	(1.05, 3.53)
Witnessed other people injured or killed	1.01	(0.64, 1.60)
Came close to being injured or killed	1.20	(0.77, 1.86)
Bullying and threats of violence	1.90 *	(1.15, 3.14)
Near-drowning	1.83 *	(1.13, 2.98)
Attempted suicide	1.62	(0.90, 2.90)
Robbery/theft	1.07	(0.69, 1.67)
Pregnancy/abortion	0.46	(0.18, 1.18)
Serious illness	1.27	(0.78, 2.07)
Death of someone close	0.71	(0.42, 1.21)
Divorce	1.13	(0.65, 1.94)
Childhood neglect	1.15	(0.87, 2.59)
Absence of a parent	1.21	(0.74, 1.98)

Notes: Reference group = None, n = 401. Nagelkerke R^2^ = 0.15. * *p* < 0.05.

**Table 8 children-11-00517-t008:** Results of the multivariable logistic regression for CPTSD: Cumulative ACEs.

Predictor	Odds Ratio	CI Interval
Sex (female)	1.24	(0.83, 1.85)
Living arrangements (One parent or other arrangements)	1.24	(0.82, 1.87)
Secure (No)	1.38	(0.90, 2.12)
Preoccupied	1.26	(0.76, 2.10)
Dismissing	1.60	(0.85, 2.99)
Fearful	1.41	(0.81, 2.47)
ACE (1)	3.22	(0.60, 10.14)
ACE (2–3)	1.44	(0.42, 4.94)
ACE (4–5)	2.34 *	(1.21, 4.54)
ACE (≥6)	1.68 *	(1.02, 2.79)

Notes: Reference group = None, n = 401. Nagelkerke R^2^ = 0.07. * *p* < 0.05.

## Data Availability

The data presented in this study are available on request from the corresponding author due to restrictions for privacy and ethical considerations and subject confidentiality.

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
