# Peer review of "Exploring Independent and Cumulative Effects of Adverse Childhood Experiences on PTSD and CPTSD a Study in Ugandan Adolescents"

_children, 2024, doi:10.3390/children11050517_

Round 1
Reviewer 1 Report
Comments and Suggestions for Authors
Dear Authors,
I enjoyed your paper and the topic that has been covered by it; I only have few suggestions to slightly adjust your paper:
1) First, In order to make its scientific soundness higher, I suggest you explain why PTSD and CPTSD are recognised as two mutually exclusive disorders in ICD-11 and yet DSM-V does not recognise CPTSD as a disorder. I suggest you also refer to the reasons why DSM-V does not recognise it and why it is important for your paper to stick with ICD-11 classification.
2) I'd also suggest widening the concept of DSO.
3) One of the limitations of your research is "usability". How can your research design be used by other researchers? How would you use it in different contexts?
Author Response
Thank you for reviewing our manuscript and for providing supportive comments. We appreciate the effort you made to improve this manuscript and are grateful for your insightful comments. The discussion below responds directly to the specific comments made by the Reviewer. The Reviewer’s comments are in bold and italicized. The authors’ comments are in normal font.
Response to Reviewer#1
1) First, In order to make its scientific soundness higher, I suggest you explain why PTSD and CPTSD are recognised as two mutually exclusive disorders in ICD-11 and yet DSM-V does not recognise CPTSD as a disorder. I suggest you also refer to the reasons why DSM-V does not recognise it and why it is important for your paper to stick with ICD-11 classification.
The requested explanation has been added to the new version of the manuscript.
2) I'd also suggest widening the concept of DSO.
The concept of DSO has been clarified in this new version of the manuscript.
3) One of the limitations of your research is "usability". How can your research design be used by other researchers? How would you use it in different contexts?
We have added the following proposal: "It is suggested that future studies analyse the independent and cumulative effects of exposure to ACEs on the mental health of adolescents in other cultural contexts.”
Reviewer 2 Report
Comments and Suggestions for Authors
I want to thank for the opportunity to review this important paper. I have some doubts regarding the methodological section.
Study group- there's a lack of information about the three cties that the participants lived. Could it be important in analysis? Has the Authors checked if the there were no statistical differences between three cities?
Line 265- "explaining the questions that the researcher was not able to"--- could the Researchers give some examples of such questions?
Did participants have an option for psychological support in case of trauma caused by questionnaires (difficult experiences/sensitive issues mentioned in the questionnaires?)
Line 279—The questionnaire names mention "Adults," but the Authors suggest that it's often used in studies on adolescents.
Author Response
Thank you for reviewing our manuscript and for providing supportive comments. We appreciate the effort you made to improve this manuscript and are grateful for your insightful comments. The discussion below responds directly to the specific comments made by the Reviewer. The Reviewer’s comments are in bold and italicized. The authors’ comments are in normal font.
Response to Reviewer#2
Study group- there's a lack of information about the three cties that the participants lived. Could it be important in analysis? Has the Authors checked if the there were no statistical differences between three cities?
Additional information on the cities where the data was collected has been provided. Thank you very much for your suggestion, but the aim of this study was to try to analyze the hypotheses of the study in a sample of adolescents from different contexts. I think your suggestion would be an excellent idea for a future study, requiring a literature review on the topic to conduct a study with this objective.
Line 265- "explaining the questions that the researcher was not able to"--- could the Researchers give some examples of such questions?
Examples of the support provided by teachers were given.
Did participants have an option for psychological support in case of trauma caused by questionnaires (difficult experiences/sensitive issues mentioned in the questionnaires?)
Yes, they had. This information was added to the manuscript.
Line 279—The questionnaire names mention "Adults," but the Authors suggest that it's often used in studies on adolescents.
As mentioned in the manuscript, the RAAS is a widely used measure in studies with adolescents from different contexts.
Reviewer 3 Report
Comments and Suggestions for Authors
Opinion on the article “Exploring Independent and Cumulative Effects of Adverse Childhood Experiences on PTSD and CPTSD. A Study in Ugandan Adolescents.”
The manuscript submitted for review is well written and contains many strengths. These include a large research sample (a sample of 401 schoolchildren) from a lower-middle-income country.
There are some ambiguities in the Conclusion section. Since the study was performed in 2012, it is interesting to see what trauma treatment assistance looks like in Uganda currently.
Line 537: “public health priority in India and Uganda” – while India? Justification needed
Author Response
Thank you for reviewing our manuscript and for providing supportive comments. We appreciate the effort you made to improve this manuscript and are grateful for your insightful comments. The discussion below responds directly to the specific comments made by the Reviewer. The Reviewer’s comments are in bold and italicized. The authors’ comments are in normal font.
Response to Reviewer#3
There are some ambiguities in the Conclusion section. Since the study was performed in 2012, it is interesting to see what trauma treatment assistance looks like in Uganda currently.
This is a limitation of the study. One of our suggestions is to carry out a new study with more up-to-date data.
Line 537: “public health priority in India and Uganda” – while India? Justification needed.
We apologize, but this is a typo. The word "India" has been removed from the manuscript.
Reviewer 4 Report
Comments and Suggestions for Authors
This paper examines the independent and cumulative effects of adverse childhood experiences (ACEs) and attachment styles on the likelihood of having a PTSD or CPTSD diagnosis in a small sample of Ugandan adolescents. Overall, the paper is interesting and sheds light on an important public health topic for an understudied population. However, I have numerous concerns as detailed below.
Major:
I have many questions about your methods and analyses.
Study Design
More details are needed on your study design. Specifically:
- How did you arrive at a sample size of 401? How many students were asked to participate? How many declined?
- How were schools selected and how were adolescents within schools selected?
- Why were only boarding schools included in your sample? How are students who attend boarding schools different from students who attend traditional schools?
- When did data collection occur? We learn in the limitations section that data were collected in 2012. Please provide dates of data collection in the Methods section.
- Did all schools complete the surveys around the same time?
- Was child assent obtained? If not, why not? This is especially troubling given the potentially distressing questions being asked of them.
Measures
Provide details on the categories for the sociodemographic measures.
It would be helpful to provide a table showing how the 20 life-threatening experiences and stressful family conditions were combined to produce the 15 grouped traumas.
You indicate that the RAAS was used to create the four attachment styles. As written, individuals who selected the midpoint are not classified, as you defined “high” as being above the midpoint, and “low” as being below the midpoint. How are individuals who selected the midpoint categorized?
Data Analysis
I have never heard of “multiple set analyses”. What is this? Please provide a reference.
Your outcomes (yes/no PTSD diagnosis, yes/no CPTSD diagnosis) are binary. Multinomial logistic regression is only appropriate if your outcome has more than two categories. Is it possible you meant multivariable logistic regression?
In the measures section, it sounds like everyone is categorized into one of the four attachment styles and this is confirmed in Table 4. Given this, it is not clear how you could have included all four attachment styles in your model without omitting one as the reference group.
Please indicate what software package was used for your analyses.
Were your models adjusted for clustering within school? If not, why not?
Given the focus on sex differences, did you test for interactions by sex or stratify your models by sex? If not, why not?
Results
Why is Table 3 stratified by sex, but Table 4 is not? Please provide attachment styles, diagnoses, and counts of ACEs by sex, along with the p-value for the comparison. Table 4 could be combined with Table 3.
The results presented in Table 5 do not look like the output of a multinomial logistic model. If you had three categories, then you would have ORs for category 2 vs. category 1 and category 3 vs. category 1. This further reinforces my belief that you conducted multivariable binary logistic regression, not multinomial logistic regression.
Table 5: What is the reference group for living arrangements? Your note in the table indicates “None” , but it’s not clear what measure that is referring to. Please explicitly state the reference group for each measure.
Table 5: In the methods section, attachment style is described as being a four-category measure. One of these (presumably secure) should be omitted as the reference group. It's not clear how you were able to fit your model with all four categories included, unless adolescents can have more than one attachment style, but that is not how this was described in the measures section.
Information on how many adolescents are in each category of the count of ACEs is needed.
The model fit for model 1 indicates a moderate relationship between the predictors and the outcome, while the model for models 2, 3, and 4 indicate a weak relationship. Please comment on this and the consequences for your study in the paper.
Were you sufficiently powered to detect differences, especially for ACES 6+?
I additionally had concerns about your intro, discussion, and conclusions:
Introduction
Lines 134-136: Rewording is sorely needed. As written, it sounds like you are identifying early pregnancy as a corporal punishment. Also, teenage pregnancy is already listed in line 133. How are girls more vulnerable to bullying and caning? More explanation is needed if this is indeed the case.
Discussion, Limitations
I don’t understand your fifth limitation. Please explain what you mean, as it’s not clear.
Additionally, an important limitation is that you only included 412 adolescents from three boarding schools. You can hardly indicate that your study is generalizable to all of Ugandan adolescents.
Conclusion
You indicate that the findings emphasize the importance of addressing social determinants of health and promoting supportive environments, yet there was no mention of these prior to this point in your paper. Your conclusion should be revised to address the topics addressed in your paper.
Minor:
Line 5: Second author’s last name is missing.
Line 14: Define PTSD and CPTSD at first use.
Line 57: Both British and American versions of “behaviours/behaviors” is used. Choose one and be consistent.
Line 89: Missing “and” between the two diagnoses (“PTSD CPTSD” should be “PTSD and CPTSD”).
Line 97: Has should be past tense (had).
Line 98: Missing “to”: “individuals exposed six or more….” Should be “individuals exposed to six or more”.
Line 122: “Countries” should be singular.
Line 125: “Children rights” is missing possessive – should be “children’s rights”.
Line 143: Prevalence rates “above 35%” are within the range indicated for PTSD (13%-67.5%). Why are these noted separately? Why not just include the four references with the two provided for the range of PTSD in Uganda?
Line 151: Missing the word “in”: “A study conducted…” should be “In a study conducted…”.
Line 183: Missing word? “The attachment system can be activated physical separation and/or danger.” Perhaps “can be activated by physical separation…” ?
Lines 195-196: I think there are missing words. “… preoccupied attachment study high level of …” Perhaps “preoccupied attachment style with a high level of …”?
Line 226: “In the present study, it was analysed the impact” – this is awkwardly phrased and should be revised.
Line 227: Sex is a sociodemographic characteristic, not a psychosocial variable.
Line 255: The correct term is “in loco parentis” not “in parents loco”.
Line 262: “Hour” should be plural.
Line 264: “… would spare one or more teachers…” – this is awkwardly phrased and should be revised.
Line 273: “The participants were asked to participants if they had been…” should be “The participants were asked if they had been…”
Line 301-302: Is PTSS (used twice) a new term or is this supposed to be PTSD? If so, it should be spelled out before using the abbreviation.
Line 337: Sex was indicated in the measures section, not gender. What was actually asked about in the survey?
Line 339: Were there only two levels of parental education (no education, primary)? Also, a closed parenthesis is missing: “education (…no education, primary, living arrangements…”
Line 347: Adverse Childhood Experiences has already been defined as ACEs, so the abbreviation should be used instead.
Line 411-412: “Regarding, only exposure to…” Regarding what? Missing a word or two…
Line 415: “…one and half more likely” – words are missing: “one and a half times more likely”.
Line 430-431: “However, it was observed some distinction” is awkwardly phrased. Please revise so it is clearer what you are saying.
Line 432-433: An additional “and” should be deleted” “… injured or killed, and physical violence, and attempted suicide…”
Line 448: I don’t understand what is meant by “occurrence of co-occurrence of risk-taking behaviours…”
Line 469: “… and exposure to higher levels of ACEs exposure…” – delete second “exposure”.
Line 537: This is the first mention of India. How does this fit into the current study?
Comments on the Quality of English LanguageAs indicated in my comments, there are numerous places where there are awkwardly-worded sentences. Please revise so it is easier to understand.
Author Response
Thank you for reviewing our manuscript and for providing supportive comments. We appreciate the effort you made to improve this manuscript and are grateful for your insightful comments. The discussion below responds directly to the specific comments made by the Reviewer. The Reviewer’s comments are in bold and italicized. The authors’ comments are in normal font.
Response to Reviewer#4
More details are needed on your study design. Specifically:How did you arrive at a sample size of 401? How many students were asked to participate? How many declined? How were schools selected and how were adolescents within schools selected? Why were only boarding schools included in your sample? How are students who attend boarding schools different from students who attend traditional schools?
Thank you very much for your questions. Additional information on the selection of schools has been added. Regarding the size of the sample, all students who were enrolled in the eighth grade were invited to participate in the study. This information has also been added to the manuscript. The role of boarding schools in Uganda's education system has also been described.
When did data collection occur? We learn in the limitations section that data were collected in 2012. Please provide dates of data collection in the Methods section. Did all schools complete the surveys around the same time?
All school that participated in the study completed the surveys within a ten days period in August 2012. This information has been added to the manuscript.
Was child assent obtained? If not, why not? This is especially troubling given the potentially distressing questions being asked of them.
Yes, consent was obtained from the students.
Provide details on the categories for the sociodemographic measures.
The requested information has been added to the new version of the manuscript.
It would be helpful to provide a table showing how the 20 life-threatening experiences and stressful family conditions were combined to produce the 15 grouped traumas.
Thank you very much for your suggestion. As requested, a new table (Table 2) has been added with the requested information.
You indicate that the RAAS was used to create the four attachment styles. As written, individuals who selected the midpoint are not classified, as you defined “high” as being above the midpoint, and “low” as being below the midpoint. How are individuals who selected the midpoint categorized?
The following clarification on this procedure has been added: “High is defined as being above the midpoint on the 5-point scale, and low as equal or below the midpoint.”
I have never heard of “multiple set analyses”. What is this? Please provide a reference. Your outcomes (yes/no PTSD diagnosis, yes/no CPTSD diagnosis) are binary. Multinomial logistic regression is only appropriate if your outcome has more than two categories. Is it possible you meant multivariable logistic regression?
We're sorry, but it was a typo. You are correct. We replaced "multiple" by "multivariate" throughout the manuscript.
In the measures section, it sounds like everyone is categorized into one of the four attachment styles and this is confirmed in Table 4. Given this, it is not clear how you could have included all four attachment styles in your model without omitting one as the reference group.
Considering that attachment style is not an ordinal measure, but a nominal/categorical one, it would not have been possible to define a reference group. The only alternative would have been to define insured vs. uninsured, with insured as the reference group. Considering that the four attachment styles were entered as predictors, we defined for style yes vs no, with no being the reference group for each attachment style.
Please indicate what software package was used for your analyses.
This information was added to the manuscript.
Were your models adjusted for clustering within school? If not, why not?
Thank you very much for your question. No, they weren't. Our aim was to analyze the objects of the study in boarding schools in different regions of the country. However, it seems like a great suggestion for a future study, requiring an in-depth literature review on differences in different regions of Uganda.
Given the focus on sex differences, did you test for interactions by sex or stratify your models by sex? If not, why not?
As mentioned earlier, the aim of the study was to test the effect of independent and cumulative exposure to ACEs in Ugandan adolescents. Sex was introduced as a predictor. It seems an excellent suggestion to test models stratified by sex, but it would require a review of sex differences in PTSD and CPTSD diagnoses, attachment styles, and other variables that were included in the study. This has been added as a limitation of the study.
Why is Table 3 stratified by sex, but Table 4 is not? Please provide attachment styles, diagnoses, and counts of ACEs by sex, along with the p-value for the comparison. Table 4 could be combined with Table 3.
Thank you very much for your suggestion. Tables 3 and 4 have been combined.
The results presented in Table 5 do not look like the output of a multinomial logistic model. If you had three categories, then you would have ORs for category 2 vs. category 1 and category 3 vs. category 1. This further reinforces my belief that you conducted multivariable binary logistic regression, not multinomial logistic regression.
As mentioned above, this error has been corrected throughout the manuscript.
Table 5: What is the reference group for living arrangements? Your note in the table indicates “None” , but it’s not clear what measure that is referring to. Please explicitly state the reference group for each measure.
As shown in Table 5, the reference group is “One parent or other arrangements”.
Table 5: In the methods section, attachment style is described as being a four-category measure. One of these (presumably secure) should be omitted as the reference group. It's not clear how you were able to fit your model with all four categories included, unless adolescents can have more than one attachment style, but that is not how this was described in the measures section.
We have defined for style yes vs no, with no being the reference group for each attachment style. We hope the information is clearer now.
Information on how many adolescents are in each category of the count of ACEs is needed.
The requested information has been provided.
The model fit for model 1 indicates a moderate relationship between the predictors and the outcome, while the model for models 2, 3, and 4 indicate a weak relationship. Please comment on this and the consequences for your study in the paper.
The following has been added to the manuscript: “Regarding the model fit for the models, Model 1 showed a moderate relationship between the predictors and the outcome, while the remaining models showed a weak relationship between the predictors and the outcomes. These results suggest that other variables, such as the initial age of exposure to ACE or coping styles, may be associated with the development of PTSD and CPTSD. Future studies could therefore include these variables as predictors of both disorders.”
Were you sufficiently powered to detect differences, especially for ACES 6+?
We haven't presented the results, but the answer is yes.
Lines 134-136: Rewording is sorely needed. As written, it sounds like you are identifying early pregnancy as a corporal punishment. Also, teenage pregnancy is already listed in line 133. How are girls more vulnerable to bullying and caning? More explanation is needed if this is indeed the case.
The sentence has been rewritten and the requested explanation has been added.
I don’t understand your fifth limitation. Please explain what you mean, as it’s not clear.
The sentence has been rewritten as follows: “Fifth, the data were collected in 2012. Thus, there may have been changes in the type and prevalence of ACEs to which adolescents are exposed nowadays.”
Additionally, an important limitation is that you only included 412 adolescents from three boarding schools. You can hardly indicate that your study is generalizable to all of Ugandan adolescents.
The following has been added. “Sixth, a convenience sample was employed in this study. Therefore, it is important to acknowledge the limitations in generalizing our findings, as our sample was drawn from specific locations and context in Uganda and may not be representative of the broader population.”
You indicate that the findings emphasize the importance of addressing social determinants of health and promoting supportive environments, yet there was no mention of these prior to this point in your paper. Your conclusion should be revised to address the topics addressed in your paper.
The conclusion has been rewritten according to your suggestions.
Line 5: Second author’s last name is missing.
The last name of Francisco, Frias, has been added.
Line 14: Define PTSD and CPTSD at first use.
The requested definition has been added.
Line 57: Both British and American versions of “behaviours/behaviors” is used. Choose one and be consistent.
This was corrected throughout the manuscript.
Line 89: Missing “and” between the two diagnoses (“PTSD CPTSD” should be “PTSD and CPTSD”).
This typo has been corrected.
Line 97: Has should be past tense (had).
This typo has been corrected.
Line 98: Missing “to”: “individuals exposed six or more….” Should be “individuals exposed to six or more”.
This typo has been corrected.
Line 122: “Countries” should be singular.
This typo has been corrected.
Line 125: “Children rights” is missing possessive – should be “children’s rights”.
This typo has been corrected.
Line 143: Prevalence rates “above 35%” are within the range indicated for PTSD (13%-67.5%). Why are these noted separately? Why not just include the four references with the two provided for the range of PTSD in Uganda?
As you suggested, we put the references together in the same sentence.
Line 151: Missing the word “in”: “A study conducted…” should be “In a study conducted…”.
This typo has been corrected.
Line 183: Missing word? “The attachment system can be activated physical separation and/or danger.” Perhaps “can be activated by physical separation…” ?
This typo has been corrected.
Lines 195-196: I think there are missing words. “… preoccupied attachment study high level of …” Perhaps “preoccupied attachment style with a high level of …”?
The sentence has been rewritten.
Line 226: “In the present study, it was analysed the impact” – this is awkwardly phrased and should be revised.
The sentence has been revised.
Line 227: Sex is a sociodemographic characteristic, not a psychosocial variable.
This has been corrected in the manuscript.
Line 255: The correct term is “in loco parentis” not “in parents loco”.
This term has been corrected.
Line 262: “Hour” should be plural.
This typo has been corrected.
Line 264: “… would spare one or more teachers…” – this is awkwardly phrased and should be revised.
This sentence has been rewritten as follows: “Headmaster of all schools were asked if they could provide some teachers to assist in answering and explaining the questions that the researcher was not able to, such as language difficulties…”
Line 273: “The participants were asked to participants if they had been…” should be “The participants were asked if they had been…”
This sentence has been revised.
Line 301-302: Is PTSS (used twice) a new term or is this supposed to be PTSD? If so, it should be spelled out before using the abbreviation.
We're very sorry, but it was a typo. It has been corrected to PTSD.
Line 337: Sex was indicated in the measures section, not gender. What was actually asked about in the survey?
The word “gender” was replaced by “sex”.
Line 339: Were there only two levels of parental education (no education, primary)? Also, a closed parenthesis is missing: “education (…no education, primary, living arrangements…”
We removed this information from the manuscript.
Line 347: Adverse Childhood Experiences has already been defined as ACEs, so the abbreviation should be used instead.
This detail has been revised.
Line 411-412: “Regarding, only exposure to…” Regarding what? Missing a word or two…
The sentence has been revised as follows. “Regarding the cumulative exposure to ACEs…”
Line 415: “…one and half more likely” – words are missing: “one and a half times more likely”.
The sentence has been recised.
Line 430-431: “However, it was observed some distinction” is awkwardly phrased. Please revise so it is clearer what you are saying.
The sentence has been revised.
Line 432-433: An additional “and” should be deleted” “… injured or killed, and physical violence, and attempted suicide…”
The sentence has been revised.
Line 448: I don’t understand what is meant by “occurrence of co-occurrence of risk-taking behaviours…”
The sentence has been revised.
Line 469: “… and exposure to higher levels of ACEs exposure…” – delete second “exposure”.
As suggested, the word has been deleted.
Line 537: This is the first mention of India. How does this fit into the current study?
We apologize, but this is a typo. The word "India" has been removed from the manuscript.
Comments on the Quality of English Language. As indicated in my comments, there are numerous places where there are awkwardly-worded sentences. Please revise so it is easier to understand.
The manuscript has been edited and revised. We hope you're pleased with the new version of the manuscript.